# Oral Corticosteroid Abuse and Self-Prescription in Italy: A Perspective from Community Pharmacists and Sales Reports before and during the COVID-19 Era

**DOI:** 10.3390/jpm13050833

**Published:** 2023-05-15

**Authors:** Emanuele Nappi, Enrico Keber, Giovanni Paoletti, Marta Casini, Carolina Carosio, Flora Romano, Nicolina Floris, Claudio Parmigiani, Carlo Salvioni, Luca Malvezzi, Francesca Puggioni, Giorgio Walter Canonica, Enrico Heffler, Corrado Giua

**Affiliations:** 1Personalized Medicine, Asthma and Allergy, IRCCS Humanitas Research Hospital, Via Manzoni 56, Rozzano, 20089 Milan, Italy; emanuele.nappi@humanitas.it (E.N.); francesca.puggioni@humanitas.it (F.P.); giorgio_walter.canonica@hunimed.eu (G.W.C.); enrico.heffler@hunimed.eu (E.H.); 2Società Italiana Farmacia Clinica (SIFAC), Viale Regina Margherita 30, 09124 Cagliari, Italy; enrico.keber@gmail.com (E.K.); floraroma_93@libero.it (F.R.); ni.26@hotmail.it (N.F.); corrado.giua@gmail.com (C.G.); 3Department of Biomedical Sciences, Humanitas University, Via Rita Levi Montalcini 4, Pieve Emanuele, 20090 Milan, Italy; marta.casini@alumni.hunimed.eu (M.C.); luca.malvezzi@humanitas.it (L.M.); 4Respiratory Disease and Lung Function Unit, Department of Medicine and Surgery, University of Parma, 43125 Parma, Italy; 5Federazione Associazione Giovani Farmacisti (Fenagifar), Via Spadolini 7, 20141 Milan, Italy; caro-c@hotmail.it; 6IQVIA™ Italia, Via Fabio Filzi 29, 20124 Milan, Italy; claudio.parmigiani@iqvia.com (C.P.); carlo.salvioni@iqvia.com (C.S.); 7Otorhinolaryngology Head & Neck Surgery Unit, IRCCS Humanitas Research Hospital, Via Manzoni 56, Rozzano, 20089 Milan, Italy; 8Otorhinolaryngology Head & Neck Surgery Unit, Casa di Cura Humanitas San Pio X, Via Francesco Nava 31, 20159 Milan, Italy

**Keywords:** oral corticosteroids, abuse, Italy, pharmacists, COVID-19

## Abstract

(1) Background: Corticosteroids are commonly used for a variety of conditions, but their use might come with significant side effects. Self-medication practices increased during the COVID-19 pandemic, potentially favoring corticosteroid misuse. Studies on this topic are lacking, thus we aim to characterize the misuse of corticosteroids in Italy through pharmacists’ perspectives and sales reports. (2) Methods: We sent to territorial pharmacists a survey that aimed to investigate corticosteroid misuse before and during the pandemic. In parallel, sales reports of the major oral corticosteroids were obtained from IQVIA. (3) Results: We found that 34.8% of clients demanded systemic corticosteroids without a valid prescription, with a rise to 43.9% during the pandemic (*p* < 0.001). Adults and patients suffering from upper airway diseases or obstructive airway diseases most frequently asked for corticosteroids without an appropriate prescription. The greatest increase after the beginning of the pandemic was seen for lung diseases. Although sales of the major oral corticosteroids decreased during the pandemic, sales of those used for COVID-19 increased. (4) Conclusions: Self-medication with corticosteroids is common and might lead to avoidable toxicities. This tendency increased during the pandemic probably because of incorrect beliefs about the inappropriate use of corticosteroids for treating COVID-19 itself. The development of shared strategies between doctors and pharmacists is essential in defining protocols guiding appropriate patient referral in order to minimize corticosteroid misuse.

## 1. Introduction

Corticosteroids are vastly prescribed in clinical practice for many health conditions such as chronic inflammatory disorders, autoimmunity, and allergies because they are potent and effective anti-inflammatory drugs able to inhibit inflammation and regulate autoimmunity [1,2,3]. Systemic therapy with corticosteroids can be administered orally, intravenously, or intramuscularly [4,5]. When they are employed for the treatment of immune-mediated disorders, a pharmacologic dose of corticosteroids is used, meaning a dose exceeding the physiologic need for glucocorticoid replacement [4]. For many inflammatory diseases, particularly for severe or refractory cases, corticosteroids might be used in the long term. For chronic use, synthetic products are preferred because of their increased potency but minimal mineralocorticoid activity [4]. The long-term use of systemic corticosteroids or therapy at high doses can result in a variety of serious side and adverse effects, including osteoporosis and related bone fractures, glaucoma and cataract, diabetes, severe infections, hypertension, hypothalamic–pituitary–adrenal axis suppression, and Cushing syndrome [4,5,6] but also neuropsychiatric adverse events at the psychological, cognitive, and/or behavioral level associated with MRI changes (widespread reduction in white matter integrity) [7,8]. Moreover, the complications of corticosteroid therapy bring about a remarkable economic burden [9]. For example, investigators of the Severe Asthma Network in Italy (SANI) formulated a real-life budget impact model which revealed that the total annual cost per severe asthma patient due to corticosteroid-related complications approximates €1957.50 [10].

The excessive and prolonged use of corticosteroids in chronic inflammatory disorders is a matter of concern among specialists. Generally, corticosteroids are applied in acute flares and are eventually tapered over time, but in many cases, chronic or repeated use is necessary to keep the disease under control. For diseases associated with a high corticosteroid burden, for example, asthma or rheumatoid arthritis, great efforts are being made to implement corticosteroid-sparing therapies and outline stewardship measures aimed at minimizing their use [11,12]. However, possibly because of a combination of their high efficacy, relatively rapid onset of action, and low cost, corticosteroids are still widely used in chronic inflammatory diseases, and many patients are overexposed to their detrimental side effects.

Patients who have experienced the outstanding anti-inflammatory benefits of this pharmaceutical class tend to self-medicate, underestimating their negative effects [13,14]. The COVID-19 pandemic has led people to increase self-medication practices. The WHO has defined self-medication as “the use of medicinal products by the consumer to treat self-recognized disorders or symptoms, or the intermittent or continued use of a medication prescribed by a physician for chronic or recurring diseases or symptoms” [15]. This phenomenon might be related to the prevention or/and treatment of COVID-19 symptoms and anxiety concerning the disease; the different lockdowns, with the consequent difficulty accessing healthcare; and, lastly, the disclosure by the media of the beneficial effect of corticosteroids against the virus [16,17,18]. It is better to keep in mind that self-medication practices come with established risks, such as incorrect self-diagnosis, choice of therapy, and dosages; excessively prolonged use; and drug interactions [15]. Nowadays, it is considered standard practice to consult the Web whenever in doubt but without necessarily having enough knowledge to evaluate the scientific validity of the information acquired [19]. The pharmacist is an important figure in providing clear, practical, and accurate answers to questions arising from patients; their direct connection with the public is essential in debunking any fake news, together with guiding the patient to seek appropriate medical assistance [20]. Two recent studies in the community pharmacy setting highlighted a high potential for abuse of certain classes of medicines in community pharmacies [21,22].

The use of glucocorticoids increased worldwide after the RECOVERY trial, which demonstrated the beneficial use of dexamethasone but only in patients receiving either invasive mechanical ventilation or oxygen therapy [23]. Particularly at the beginning of the pandemic, specific treatment options for severe COVID-19 were limited [24,25]. In Italy, the use of corticosteroids for severe COVID-19 was supported by the Italian Agency of Drugs (AIFA) starting in October 2020 [26].

To date, the literature on the misuse of systemic corticosteroids in Italy is scarce and unclear. This study is the result of an investigation conducted by the Italian Society of Clinical Pharmacy in collaboration with the Department of Biomedical Sciences of Humanitas University, and it is aimed at defining the objective consumption of some of the most common oral corticosteroid molecules in the pre- and post-pandemic periods; understand, according to the perception of pharmacists, the clinical aspects underlying the use of such therapies; and correlate the perceived consumption and clinical data to define the actual impact of the misunderstood problem and abuse of this drug class.

## 2. Materials and Methods

### 2.1. Study Design

The study was developed into two distinct parts:
Data were collected through the development of an online survey distributed to Italian pharmacists with the goal of investigating the pharmacists’ perceptions of systemic corticosteroid use and abuse.Sales reports of the major systemic oral corticosteroids in Italian community pharmacies in the pre-pandemic period from 1 March 2019 to 20 February 2020 and during the pandemic period from 1 March 2020 to 28 February 2021 were acquired from IQVIA™, a global provider of information, innovative technologies, and clinical research services for the industries of health information technology and clinical research. These data were gathered to put into context the perceptions obtained through the field investigation.


### 2.2. Study Population

To obtain the target population, the sample size was calculated using a single population proportion formula by Raosoft (2004), the Raosoft Sample Size Calculator [27], with a confidence interval of 95% and a marginal error of 5.61%. The sample size considered was 19,669 pharmacies on the Italian territory [28]. The survey was sent by email to 400 Italian pharmacists using the network of members of the “Società Italiana di Farmacia Clinica” (SIFAC) and “Federazione Nazionale Associazioni Giovani Farmacisti” (Fenagifar). Participation was entirely voluntary; moreover, in order to ensure a homogenous distribution in the Italian National Territory, pharmacists were contacted individually.

### 2.3. Questionnaire

The first version of the questionnaire was drawn up by a group of researchers from the Italian Society of Clinical Pharmacy and the Department of Biomedical Sciences of Humanitas University. The questionnaire consisted of five questions. A team of clinical pharmacists (SGCP) evaluated the usefulness and comprehension of the survey by providing written feedback to the research team. The final version of the survey was drawn up on the basis of the feedback. The web-based survey was created using Google^®^ Forms to collect data. The survey was available from 1 March 2022 to 15 April 2022. The questionnaire in Italian and English is available in the Appendix A.

### 2.4. Ethical Statement

This study was conducted in conformity with the Declaration of Helsinki. Electronic informed consent was obtained from each participant before the start of the investigation.

### 2.5. Data Analysis

Q1-3-4: Questions 1, 3, and 4 investigated pharmacists’ perceptions of corticosteroid abuse in quantitative terms, considering all corticosteroid users (Q1) or subdividing them according to the type of patient (“chronic user” versus “first time user”; Q3) and according to the reason for the lack of an appropriate prescription (“Emergency use” versus “Difficulty in getting a prescription” versus “Forgotten prescription”; Q4). Pharmacists were asked to provide distinct answers as regards the pre-pandemic period and the pandemic period. Answers were collected as numerical ranges (0–10, 11–20, 21–30, 31–40, 41–50, 51–60, 61–70, 71–80, 81–90, and 91–100 out of 100 patients). A Student’s t-test was used to assess statistical significance between answers referring to the pre- and post-pandemic periods.

To improve clarity of data analysis and representation, original answers were converted into numerical data (respectively, 0.1, 0.2, 0.3, 0.4, 0.5, 0.6, 0.7, 0.8, 0.9, and 1). To evaluate the effect of the pandemic on corticosteroid abuse, paired answers from each pharmacist were considered (from before and after the pandemic). Owing to the large sample size, statistical significance was assessed with a Paired Samples t-test.

Q2-5: Questions 2 and 5 investigated pharmacists’ perceptions of corticosteroid abuse according to patients’ age group (Q2) and underlying medical condition (Q5). Pharmacists were asked to provide distinct answers as regards the pre-pandemic period and the pandemic period. Answers about the frequency of corticosteroid abuse (“never”, “sometimes”, “often”, “very often”, and “always”) were collected. The pharmacists’ answers were grouped into “frequent” (including “often”, “very often”, and “always”) or “not frequent” (including “never” and “sometimes”). To evaluate the effect of the pandemic on corticosteroid abuse, paired answers from each pharmacist were considered (from before and after the pandemic). Statistical significance was then assessed with McNemar’s test.

Data analysis was conducted with IBM^®^ SPSS Software for Macintosh, Version 26.0 (IBM Corp., Armonk, NY, USA). Statistical significance was defined as *p* < 0.05.

## 3. Results

### 3.1. Participants

A total of 376 pharmacists were recruited. Demographic and occupational characteristics were assessed and are summarized in Table 1.

The complete list of questions (Q1–5) as well as original answers provided by pharmacists are reported in the Appendix A.

Q1. According to pharmacists, a high fraction of clients demanded oral corticosteroids without an appropriate prescription, both before (mean 34.8%, SD 23%) and after the pandemic (mean 43.9%, SD 26.2%), as reported in Table 1. A comparison of the answers from the pre- and post-pandemic periods demonstrated a statistically significant increase in the fraction of clients demanding oral corticosteroids without an appropriate prescription (Difference of Means 9.1%; 95% Confidence Interval 7.7 to 10.6%; *p*-value < 0.001) see Figure 1.

Q2. This question investigated the distribution of systemic corticosteroid requests without a valid prescription in different age groups. As shown in Table 2 and Figure 2, adults (36–60 years-old) were the age group that most frequently asked for oral corticosteroids without a valid prescription. In particular, 62.5% (235/376) of pharmacists reported that adults frequently (including the answers “often”, “very often” and “always”) asked for oral corticosteroids without a valid prescription both before and after the pandemic, and an additional 6.9% (26/376) changed from “not frequent” (including the answers “never” and “sometimes”) before the pandemic to “frequent” after the pandemic. In contrast, it was less frequent for clients at the extremes of age to ask for systemic corticosteroids without a valid prescription (Table 2). A statistically significant increase in the frequency of systemic corticosteroid requests without an appropriate prescription between the two time periods has been noted in almost all age groups.

Q3. According to the pharmacists, most clients without a valid prescription were chronic users, both before (mean answer 66.0%, SD 20.7%) and after the pandemic (mean answer 63.1%, standard deviation 21.2%). The data are summarized in Figure 3. A comparison of answers from the pre- and post-pandemic periods demonstrated a statistically significant increase in the fraction of first-time users demanding oral corticosteroids without an appropriate prescription (Difference of Means 2.8%; 95% Confidence Interval 1.4 to 4.3%; *p*-value < 0.001), paralleled by a small but statistically significant decrease in the fraction of chronic users (Difference of Means = −2.9%; 95% Confidence Interval −4.4 to −1.4%; *p*-value < 0.001).

Q4. When investigating the reasons why clients asked for oral corticosteroids without an appropriate prescription, it appeared that “emergency use” and “difficulty in getting the prescription” are the most frequent reasons. A statistically significant increase in the fraction of clients encountering difficulties in obtaining the prescription was noted (pre-pandemic mean: 38.7%, SD: 19.1%; post-pandemic mean: 45.6%, SD: 21.1%; Difference of Means: 6.9%; 95% Confidence Interval 4.0% to 9.8%; *p*-value < 0.001). This was paralleled by a significant decrease in the fraction of patients who forgot their prescription (pre-pandemic mean: 29.6%, SD 18.1%; post-pandemic mean: 24.0%, SD 16.2%; Difference of Means: 5.6%; 95% Confidence Interval: −8.0% to −3.1%; *p*-value < 0.001). In contrast, no significant variation in the fraction of clients who required oral corticosteroids for emergency use was observed (pre-pandemic mean: 39.5%, SD 21.7%; post-pandemic mean: 39.3%, SD 19.9%; Difference of Means: −0.2%; 95% Confidence Interval −3.2% to 2.7%; *p*-value 0.875). The data are represented in Figure 4.

Q5. Pharmacists were asked to specify the frequency of corticosteroid requests without a valid prescription in distinct medical conditions. According to the pharmacists, before the pandemic, upper airway and obstructive airway diseases were the conditions for which systemic corticosteroids were requested without a valid prescription more often than not (respectively, 75.3% and 68.1%), whereas such requests were the least frequent for rheumatological disorders (23.1%), as shown in Table 3 and Figure 5. Following the pandemic, a statistically significant increase in the number of patients asking for corticosteroids without a valid prescription for sore throat, rheumatological diseases, obstructive airway disease, pulmonary diseases, and cutaneous diseases was noted by the pharmacists. In absolute terms, the most notable increase was noted for pulmonary conditions, whereby 88 pharmacists (23.4%) changed their answer from “not frequent” before the pandemic to “frequent” after the pandemic.

### 3.2. IQVIA™ Sales Reports

Sales reports of the major oral corticosteroids dispensed in Italian territorial pharmacists in the pre-pandemic period from 1 March 2019 to 29 February 2020 and during the pandemic from 1 March 2020 to 28 February 2021 were acquired from IQVIA™ and are reported in Table 4. Total sales declined between the two time periods analyzed (from 26,142,619 to 23,370,484). Looking at specific molecules, betamethasone, deflazacort, and methylprednisolone sales have decreased, whereas dexamethasone and prednisone sales have increased.

## 4. Discussion

Corticosteroids are highly effective in the treatment of numerous health conditions, but their use might come with significant side effects on various systems, including the musculoskeletal, cardiovascular, endocrinologic, infectious, ocular, and neuropsychiatric systems [4,5,6,7]. We are concerned that the greater tendency of the general population to self-medicate, together with the restrained access to timely medical advice due to the COVID-19 pandemic, might have fostered the misuse and abuse of corticosteroids, particularly at the territorial level. Pharmacists are frequently the first healthcare workers encountered by patients, playing a critical role in providing advice and guiding patients to seek appropriate assistance [29]. Therefore, we developed a questionnaire for pharmacists aimed at investigating and characterizing the misuse of corticosteroids at the territorial level both before and during the pandemic. The questionnaire was completed by 376 Italian pharmacists with various academic backgrounds and who practiced in urban (73.4%) and rural (26.4%) areas, as illustrated in Table 1.

According to pharmacists, the percentage of patients who asked for systemic corticosteroids without a valid prescription (without a prescription or with an expired prescription) approximated 34.8% before the pandemic, and this fraction rose to 43.9% after the beginning of the pandemic (*p*-value < 0.001). Furthermore, research conducted by Perelló et al. [21] revealed that failure to present a prescription perpetuates drug misuse in 58.6% of cases (although the investigators of this study assessed the misuse of other drug classes). Community pharmacists can play a key role in managing medication misuse [30]. The development of shared strategies between specialist doctors and pharmacists is essential for defining protocols guiding appropriate patient referrals in high-risk situations. Innovative approaches aimed at intercepting and minimizing the effects caused by drug misuse should be developed, as they could foster better management of this issue, for example, screening activities and dedicated patient services.

Interestingly, most “corticosteroid abusers” were chronic users, suggesting that patients who already benefitted from these drugs might continue their use without seeking further medical advice, possibly underestimating their potential negative effects. Indeed, some of the immediate adverse effects of steroids (e.g., insulin resistance, hypertension, loss of bone mineral density) may be unnoticed by patients until complications of prolonged therapy develop. Patients in this group—“chronic corticosteroid abusers”—are likely to be exposed to a very high risk of adverse effects. It would be relevant to assess whether they could benefit from other therapeutical strategies that would reduce the burden of corticosteroid-related complications.

During the pandemic, a statistically significant increase in the fraction of “first-time” users asking for corticosteroids without an appropriate prescription was noted, paralleled by a decrease in the fraction of chronic users. This change may be accounted for by the acknowledgment that steroids can be beneficial in severe COVID-19 cases [17] and the greater difficulty in accessing healthcare facilities during the COVID-19 pandemic. Importantly, short courses (less than 30 days) of systemic corticosteroids increase the risk of serious adverse effects such as sepsis, venous thromboembolisms, and fractures [31].

When looking at the various medical conditions, it appears that individuals affected by upper airway or obstructive airway diseases were the ones that most commonly asked for systemic corticosteroids without an appropriate prescription (as represented in Table 3 and Figure 5). In particular, over 50% of pharmacists said that patients affected by these pathologies frequently asked for systemic corticosteroids without a valid prescription. Asthma and chronic obstructive pulmonary disease (COPD) are the major obstructive airway diseases and probably account for most patients in this category, as systemic corticosteroids may be applied in acute flares of these conditions. The administration of systemic corticosteroids in this setting should occur under medical guidance; nevertheless, our results suggest that many patients affected by obstructive airway diseases tend to self-medicate, seeking systemic corticosteroids without a valid prescription, thus probably without receiving previous medical advice. The overuse of systemic corticosteroids in asthma is a matter of concern among specialists and should be minimized as much as possible [9,32,33]. However, real-life evidence shows that up to 9% of patients with asthma receive systemic corticosteroids for more than 30 days per year despite the availability of safer therapeutical options; indeed, approximately one-third of these patients were not in treatment step 5, suggesting that other treatment options were not adequately assessed [34]. Even the occasional use of systemic corticosteroids is associated with worse outcomes and should be minimized as much as possible: a large longitudinal cohort study in asthma patients from the UK showed that intermittent systemic corticosteroid use is associated with a relevant increase in the risk of related adverse effects, and almost one-third of intermittent systemic corticosteroid users will eventually develop a frequent pattern of use [35].

During the pandemic, an increasing trend in the demand for corticosteroids without a valid prescription has been noted for most medical conditions, particularly for lung diseases (Table 3). Likely, COVID-19 was one of the pulmonary conditions for which individuals asked for steroids without a prescription, as the media disclosed that these drugs may be beneficial for COVID-19. However, steroids have been proven to be beneficial only in severe COVID-19 cases [17], and the pharmacists who replied to our questionnaire were dispensing drugs to outpatients likely without severe disease. It is possible that among the general population, there has also been a tendency toward the misuse of corticosteroids in mild COVID-19 cases, pushed by the fear of severe diseases; this would explain why more outpatients requested systemic corticosteroids for pulmonary conditions without a prescription during the pandemic period.

Adults most frequently asked for oral corticosteroids without an appropriate prescription, and this tendency decreased going toward the extremes of ages. During the pandemic, an increase in inappropriate corticosteroid requests emerged across all age groups (Table 2 and Figure 2). In most cases, this increase was statistically significant, with a few exceptions that are mainly related to those groups of patients (e.g., adult patients, patients suffering from upper airway disease) for which corticosteroid abuse was already extremely frequent in the pre-pandemic period. These data emphasize the great strain that the COVID-19 pandemic put on the Italian National Health System. Indeed, a relevant fraction of clients without valid prescriptions complained about the hardship of obtaining a prescription (on average 38.7% before the pandemic), and this figure significantly increased during the pandemic (on average 45.6%; *p*-value < 0.001), reflecting the greater difficulties in receiving medical assistance throughout this period (data are shown in Figure 4).

It is possible that oral corticosteroid misuse will decrease to pre-pandemic figures as the COVID-19 pandemic wanes. Nevertheless, our results suggest that corticosteroid misuse was already highly prevalent in the pre-pandemic period, and it is unlikely that this issue will improve without any specific intervention. Moreover, many self-medicated individuals might have only experienced the positive anti-inflammatory effects of the drugs without any apparent side effect, making them prone to perpetuate self-medication with oral corticosteroids in the future.

Despite the relative increase in the perceived fraction of clients asking for corticosteroids without an appropriate prescription, reports from IQVIA™ (Table 4) show that the total sales of the most commonly used oral corticosteroids decreased between February 2020 and February 2021. This datum may suggest that the COVID-19 pandemic and consequent public lockdowns reduced overall sales for territorial pharmacies. However, when looking at the specific molecules, sales of dexamethasone (the synthetic corticosteroid indicated for the treatment of severe COVID-19 cases [17]) increased, and sales of prednisone and methylprednisolone (two molecules frequently used to treat lower respiratory symptoms, as well as many other conditions) remained nearly stable.

To conclude, up to 1/3 of pharmacists’ clients asked for systemic corticosteroids without a valid prescription. These individuals might be misusing oral corticosteroids, bringing about dangerous side effects. Moreover, the fraction of oral corticosteroid misusers increased once the COVID-19 pandemic began. Most misusers were chronic corticosteroid users, implying that the potential negative effects related to this therapy might be even more dramatic. The most significant increase in inappropriate corticosteroid requests during the pandemic has been noted for pulmonary pathologies, suggesting a relevant misuse of systemic corticosteroids in mild COVID-19 outpatients as well. It is understandable that during the pandemic, a higher strain on the public health system resulted in a greater tendency to ask for systemic corticosteroids without a valid prescription, but it is also of the utmost importance that as we return to normality (with the end of the state of emergency caused by the pandemic), patients return to seek medical attention before starting or continuing these medications outside of the prescription limits. It is important to underline that the strength of our conclusions may be limited by the retrospective nature of this study and by the fact that we investigated corticosteroid misuse indirectly through pharmacists’ perceptions. Moreover, our study only assessed the Italian situation; it would be interesting to investigate whether the impact of the COVID-19 pandemic on corticosteroid misuse differed in other countries with diverse public health policies, as it is possible that differences in lockdown length and strictness affected the degree of drug misuse in the general population. Despite these limitations, our study suggests that self-medication practices with corticosteroids are widespread in Italy and that the COVID-19-related healthcare emergency worsened the situation. It is critical that all patients seek appropriate and continuous medical assistance when taking medications with significant side effects (such as corticosteroids). It is our concern that the burden of side effects associated with these medications can be even greater in the context of self-medication practices.

## Figures and Tables

**Figure 1 jpm-13-00833-f001:**
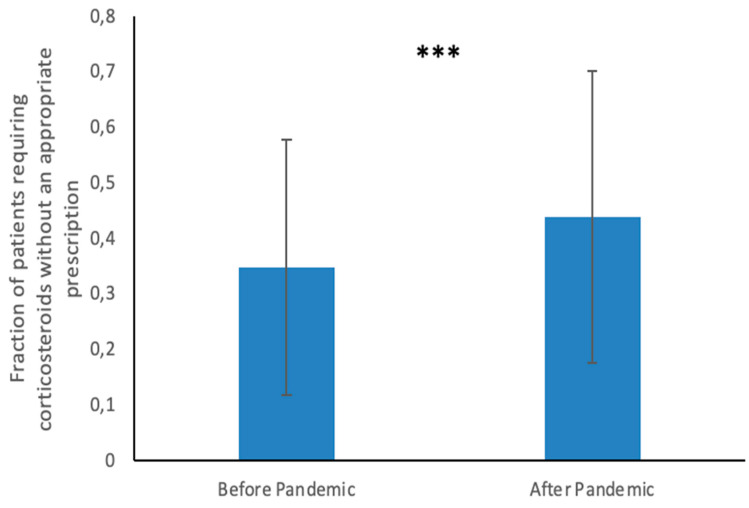
Perceptions of systemic corticosteroid abuse by pharmacists. This graph shows pharmacists’ answers to question 1 regarding the fraction of patients who required oral corticosteroids without an appropriate prescription in the pre-pandemic and post-pandemic periods. *** indicates a *p*-value < 0.001.

**Figure 2 jpm-13-00833-f002:**
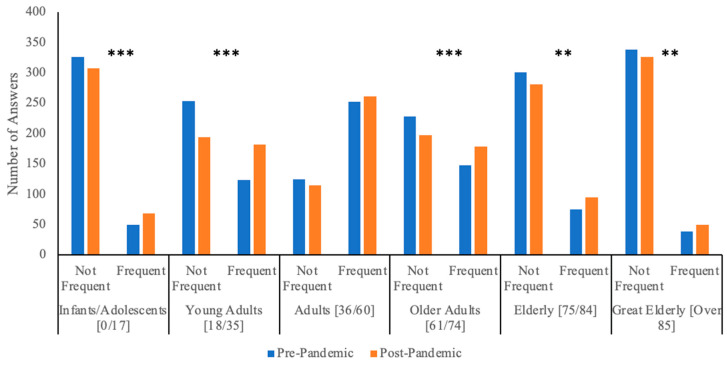
Age distribution of clients that requested systemic corticosteroids without a valid prescription. This graph shows pharmacists’ answers to question 2 regarding the frequency of corticosteroid requests without an appropriate prescription in patients of different age groups. ** indicates a *p*-value < 0.01; *** indicates a *p*-value < 0.001.

**Figure 3 jpm-13-00833-f003:**
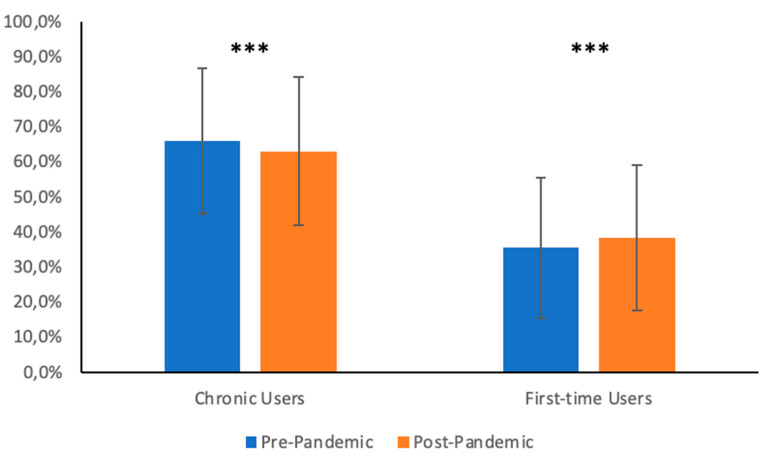
Summary of answers to question 3, divides the distribution of clients requesting systemic corticosteroids without a valid prescription into first-time users and chronic users, according to pharmacists. *** indicates a *p*-value < 0.001.

**Figure 4 jpm-13-00833-f004:**
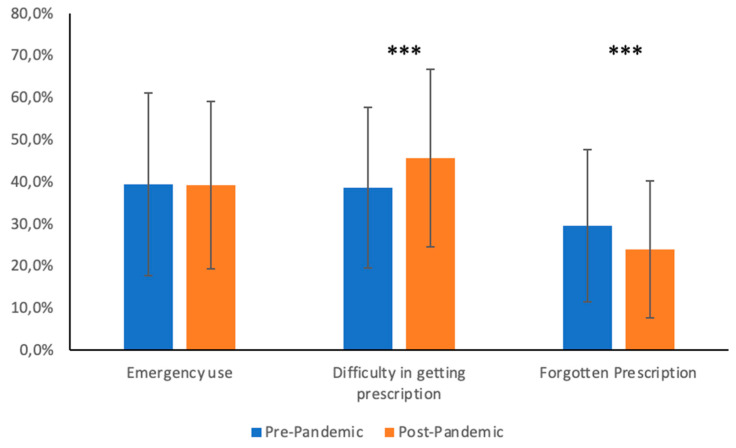
Summary of answers to question 4 showing pharmacists’ perceptions of the reasons why clients requested systemic corticosteroids without a valid prescription. *** indicates a *p*-value < 0.001.

**Figure 5 jpm-13-00833-f005:**
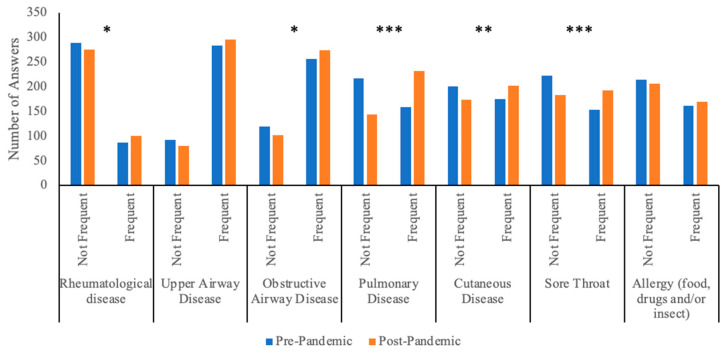
Summary of answers to question 5 showing pharmacists’ perceptions of the frequency of corticosteroid abuse in different clinical conditions. * indicates a *p*-value < 0.05; ** indicates a *p*-value < 0.01; *** indicates a *p*-value < 0.001.

**Table 1 jpm-13-00833-t001:** Demographic and occupational variables of the 376 pharmacists included in the study.

Variable	Secondary Variable	Result (n%)
Kind of pharmacy	Urban	276 (73.4%)
	Rural	100 (26.6%)
Study degree	Degree	244 (64.9%)
	Master	111(29.5%)
	Residency/Major	17 (4.5%)
	PhD	4 (1.1%)
Years since university graduation	<5 years	88 (23.4%)
	5–10 years	143 (38%)
	11–20 years	100 (26.6%)
	>20 years	45 (12%)

**Table 2 jpm-13-00833-t002:** Age distribution of clients that requested systemic corticosteroids without a valid prescription. This table shows pharmacists’ answers to question 2. Differences between the pre- and post-pandemic eras were assessed with McNemar’s test.

Age Group	Unchanged (Frequent)	Unchanged (Not Frequent)	From Not Frequent Before to Frequent After Pandemic	From Frequent Before to Not Frequent After Pandemic	*p*-Value
Infants and adolescents [0–17]	46 (12.23%)	304 (80.85%)	22 (5.85%)	4 (1.06%)	**<0.001**
Young adults [18–35]	109 (28.99%)	180 (47.87%)	73 (19.41%)	14 (3.72%)	**<0.001**
Adults [36–60]	235 (62.5%)	98 (26.06%)	26 (6.91%)	17 (4.52%)	0.17
Older adults [61–74]	134 (35.64%)	183 (48.67%)	45 (11.97%)	14 (3.72%)	**<0.001**
Elderly [75–84]	62 (16.49%)	268 (71.28%)	33 (8.78%)	13 (3.46%)	**0.003**
Great elderly [Over 85]	36 (9.57%)	324 (86.17%)	14 (3.72%)	2 (0.53%)	**0.003**

**Table 3 jpm-13-00833-t003:** Summary of answers to question 5 showing pharmacists’ perceptions of the pathologies that affected clients asking for systemic corticosteroids without a valid prescription. Statistical significance was assessed with McNemar’s test.

Underlying Medical Condition	Unchanged (Frequent)	Unchanged (Not Frequent)	From Not Frequent before Pandemic to Frequent after Pandemic	From Frequent before Pandemic to Not Frequent after Pandemic	*p*-Value
Rheumatological disease	70 (18.62%)	258 (68.62%)	31 (8.24%)	17 (4.52%)	**0.043**
Upper airway disease	264 (70.21%)	61 (16.22%)	32 (8.51%)	19 (5.05%)	0.069
Obstructive airway disease	234 (62.23%)	80 (21.28%)	40 (10.64%)	22 (5.85%)	**0.022**
Pulmonary disease	144 (38.3%)	129 (34.31%)	88 (23.4%)	15 (3.99%)	**<0.001**
Cutaneous disease	149 (39.63%)	148 (39.36%)	53 (14.1%)	26 (6.91%)	**0.002**
Sore throat	140 (37.23%)	170 (45.21%)	53 (14.1%)	13 (3.46%)	**<0.001**

**Table 4 jpm-13-00833-t004:** Sales data about the major oral corticosteroids in Italian territorial pharmacies, provided by IQVIATM, are reported. Both the pre-pandemic and the post-pandemic periods have been analyzed.

Molecule	March 2019–February 2020 (N. of Packs)	March 2020–February 2021 (N. of Packs)
Betamethasone	10,988,236	8,055,729
Deflazacort	755,338	637,367
Dexamethasone	1,304,043	1,553,025
Methylprednisolone	2,955,514	2,927,986
Prednisone	10,139,488	10,196,377
Total	26,142,619	23,370,484

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
