# Peer review of "Oral Corticosteroid Abuse and Self-Prescription in Italy: A Perspective from Community Pharmacists and Sales Reports before and during the COVID-19 Era"

_jpm, 2023, doi:10.3390/jpm13050833_

Round 1

Reviewer 1 Report

The manuscript "Oral corticosteroids abuse and self-prescription in Italy: a perspective from community pharmacists and sales reports before and during COVID era" is need some suggestions as follows:

1. The rationale of research must be included in the last section of introduction with hypothesis.

2. Remove the background lines from the Figures.

3. The caption of table must be above inserted the table 2. 

4. Include these supportive references in introduction section fist or second.

a."Antiviral activities of 4H-chromen-4-one scaffold-containing flavonoids against SARS–CoV–2 using computational and in vitro approaches." Journal of Molecular Liquids 353 (2022): 118775.

b. "Assessment of antiviral potencies of cannabinoids against SARS-CoV-2 using computational and in vitro approaches." International journal of biological macromolecules 168 (2021): 474-485.

Minor editing of English language is required

Author Response

Dear reviewer 1,

Thank you very much for your time and for your valuable feedback. We updated our manuscript according to your suggestions as follows:

  1. We clarified the aims of our study at the end of the introduction (rows 106-112).
  2. We removed background from all graphs.
  3. We placed the caption of Table 2 above Table 2 (rows 208-210).
  4. We included the two references that you proposed in the introduction, you can find it as ref. 24 and 25 in the new version of the manuscript.

We also performed English language corrections.

Reviewer 2 Report

With their survey on corticosteroid abuse before and after the COVID19-pandemic, the authors addressed a current medical and political issue that will hopefully lead to increased awareness and countermeasures to be taken. The greatest limitation of this work is that it is mostly based on the subjective perception of pharmacists, but the authors have addressed this at the end of their discussion. Another limitation they may add is that the study is restricted to Italy and potentially discuss how their results might compare to other countries with stricter and less strict lockdown measures. I would also suggest to discuss whether the identified increase in corticosteroid misuse will decrease again now that the pandemic is over. Other than that, I find the discussion to be comprehensive and addressing of all major points. The other sections of the manuscript also read well, but a few minor points should be addressed:

-Lines 55-61: Cushing-syndrome should also be mentioned

-The description for Table 2 has been displaced

-Add a brief description of what IQVIA is

Overall, the quality of English language is fine, considering that the authors are likely not native speakers. However:

-Lines 85-87: The sentence should be rephrased, as "...come with few established risks,..." has actually the opposite meaning of what the authors supposedly intended to say. Furthermore, "It's..." must be spelled out as "It is..."

-Typographical or spelling errors in lines 125 and 171

-Line 397: The sentence needs to be rephrased, as it currently portrays a false meaning.

Author Response

Dear reviewer 2,

Thank you very much for your time and for your valuable feedback. We updated our manuscript according to your suggestions as follows:

  1. We included among the limitations of our study the fact that we were able to acquire data only from the Italian territory, and that it would be interesting to assess whether there are differences in countries with different lockdown measures (rows 462-466).
  2. We discussed the possible evolutions of corticosteroid misuse once the COVID-19 pandemic will wane (rows 431-437).
  3. We included Cushing syndrome among the long-term side effects of oral corticosteroid (row 58)
  4. We positioned properly the caption of Table 2 (rows 208-210)
  5. We specified what IQVIATM is (rows 122-124).
  6. We performed the English language corrections that you suggested. Thank you very much for pointing them out.

Round 2

Reviewer 1 Report

This revised manuscript is suitable for further process in journal.